# Influence of Carbon Nanotubes Concentration on Mechanical and Electrical Properties of Poly(styrene-co-acrylonitrile) Composite Yarns Electrospun

**DOI:** 10.3390/polym13213655

**Published:** 2021-10-23

**Authors:** Rubén Caro-Briones, Blanca Estela García-Pérez, Eduardo San Martín-Martínez, Héctor Báez-Medina, Irlanda Grisel Cruz-Reyes, José Manuel del Río, Hugo Martínez-Gutiérrez, Mónica Corea

**Affiliations:** 1Escuela Superior de Ingeniería Química e Industrias Extractivas, Instituto Politécnico Nacional, Av. Luis Enrique Erro S/N, Unidad Profesional Adolfo López Mateos, Zacatenco, Alcaldía Gustavo A. Madero, Ciudad de México C.P. 07738, Mexico; rbcrbr10@gmail.com; 2Escuela Nacional de Ciencias Biológicas, Instituto Politécnico Nacional, Unidad Profesional Lázaro Cárdenas Prolongación de Carpio y Plan de Ayala S/N Col. Santo Tomas, Alcaldía Miguel Hidalgo, Ciudad de México C.P. 11340, Mexico; abrilestela@hotmail.com (B.E.G.-P.); irlandahu@hotmail.com (I.G.C.-R.); 3Centro de Investigación en Ciencia Aplicada y Tecnología Avanzada, Instituto Politécnico Nacional, Calzada Legaria No. 694 Col. Irrigación, Alcaldía Miguel Hidalgo, Ciudad de México C.P. 11500, Mexico; sanmartinedu@hotmail.com; 4Centro de Investigación en Computación, Instituto Politécnico Nacional, Av. Juan de Dios Bátiz, Esq. Miguel Othón de Mendizábal, Col. Nueva Industrial Vallejo, Alcaldía Gustavo A. Madero, Ciudad de México C.P. 07738, Mexico; hebame@gmail.com; 5Departamento en Ingeniería en Metalurgia y Materiales, ESIQIE, Instituto Politécnico Nacional. Av. Luis Enrique Erro S/N, Unidad Profesional Adolfo López Mateos, Zacatenco, Alcaldía Gustavo A. Madero, Ciudad de México C.P. 07738, Mexico; jm.delrio.garcia@gmail.com; 6Centro de Nanociencias y Micro y Nanotecnologías, Instituto Politécnico Nacional, Av. Luis Enrique Erro S/N, Unidad Profesional Adolfo López Mateos, Zacatenco, Alcaldía Gustavo A. Madero, Ciudad de México C.P. 07738, Mexico

**Keywords:** polymer composite, nanotubes, polymeric yarns, artificial muscle, mechanical properties, electrical properties

## Abstract

In this work, the influence of carbon nanotubes (CNTs) content on the mechanical and electrical properties of four series of polymeric matrix were made and their cytotoxicity on cells was evaluated to consider their use as a possible artificial muscle. For that, polymer composite yarns were electrospun using polymeric solutions at 10 wt.%. of poly(styrene-co-acrylonitrile) P(S:AN) and P(S:AN-acrylic acid) P(S:AN-AA) at several monomeric concentrations, namely 0:100, 20:80, 40:60, 50:50 (wt.%:wt.%), and 1 wt.% of AA. Carbon nanotubes (CNTs) were added to the polymeric solutions at two concentrations, 0.5 and 1.0 wt.%. PMCs yarns were collected using a blade collector. Mechanical and electrical properties of polymeric yarns indicated a dependence of CNTs content into yarns. Three areas could be found in fibers: CNTs bundles zones, distributed and aligned CNTs zones, and polymer-only zones. PMCs yarns with 0.5 wt.% CNTs concentration were found with a homogenous nanotube dispersion and axial alignment in polymeric yarn, ensuring load transfer on the polymeric matrix to CNTs, increasing the elastic modulus up to 27 MPa, and a maximum electrical current of 1.8 mA due to a good polymer–nanotube interaction.

## 1. Introduction

The polymer-matrix composite (PMC) is defined as the combination of two or more materials, where their final physical and chemical properties result in better properties than their individual components [1,2]. The continuous phase corresponds to a polymeric matrix with different physical structures for example, films, yarns or fibers and bulk [3,4,5,6] and a dispersed phase composed of nanofillers, such as nanoparticles, nanofibers, nanocrystals, nanoclays, nanorods, or nanotubes [7,8,9,10,11].

The PMCs have attracted great interest due to their multifunctional structure and properties. It has been shown that the addition of nanofillers in the polymeric matrix affects their physical structures and improves mechanical, electrical, chemical, thermal, biological, and optical properties that can be tailored to certain applications. For example, they can be used as a biomimetic actuator that provides elongation or contraction through electrical pulses applied along the composite polymer [12]; for temperature and gas sensing where polymeric functional groups play a transducer role in specific environments [13]; in microelectronic applications due to their good dielectric and heat transfer properties [14]; and for electromagnetic interference shielding because of their good absorption and reflection of radiation for electronic protection [15]. PMCs have also had a great impact in medical applications, such as extracellular matrices (ECM) for cell proliferation, scaffolds for healing wounds, as carriers and drug delivery for long periods of treatment, artificial muscles, and prosthesis [14,15]. PMCs are also used for their durability, low cost, and easy manufacturing methods [16,17].

A nanomaterial that has been reported as an interesting reinforcement is the carbon nanotubes (CNTs) of one-dimensional structure because of their good electrical, mechanical, and thermal properties [18]. In addition, their anisotropic geometry makes them suitable for reinforcing polymeric fibers and turning them into oriented polymeric systems capable of mimicking the structure of natural materials, such as bone, organs and muscles [19].

The CNTs-PMC processing involves mixing and dispersing CNTs into a low-viscosity solvent or direct blending into polymer melts. Nevertheless, because their nano-size and large length:diameter ratio, difficulties remain in dispersing them in the polymeric matrix, especially at high loadings (≥5 wt.%) [20,21]. There are some techniques to improve the dispersion of CNTs, such as sonication, the use of surfactants, high shear mixing, and chemical functionalization [22,23]. However, some of these techniques are ineffective in unbundling CNTs and are also destructive, because they shorten the length of the nanotubes, disrupt pristine sp^2^ hybridization, and impair their intrinsic properties. Dispersion efficiency is highly dependent on the choice of chemistry and viscosity of the polymeric matrix and the mixing method [19,24]. Some works have reported good CNTs dispersion in polyvinyl alcohol (PVA) at concentrations ≤1 wt.% [25].

Nowadays, artificial muscles and organs have been improved in biomedical engineering using PMCs, but it is still necessary to develop materials with better mechanical, electrical, and biocompatibility properties [26]. Polyacrylonitrile (PAN) is an acrylic polymer with excellent chemical, electrical, mechanical, and thermal properties [27,28]. PAN fibers and yarns have been widely used in electronic devices, biomedical devices and prosthetics, energy storage, and membranes because of their toughness and flexibility [29,30]. The intrinsic properties of this polymer have been enhanced by reinforcement with carbon nanofillers, such as nanotubes, graphene, or nanoparticles, while polystyrene (PS) is a thermoplastic polymer with good chemical inertness, low density, high tensile strength and Young’s modulus, and easy processability [31]. In a previous work, the synthesis of a poly(styrene-co-acrylonitrile) copolymer was reported. The results revealed a synergy between these polymers such as Young´s modulus, semiconducting properties, and non-cytotoxic effects [32]. The use of this copolymer as matrix and CNTs as dispersed phase promise a good electrical performance for potential use as possible artificial muscles.

Several works have been found about the effects of CNTs as nanofillers on mechanical, thermal, and electrical properties in these neat polymers. For example, Chae and co-workers embedded carbon nanotubes into polyacrylonitrile (PAN) fibers. Their results showed that fibers toughness improved up to 230% using 5 wt.% of CNTs [33]. Wang and Murthy also used fibers of PAN reinforced with CNTs and studied the role of orientation of them into fibers. The results showed that aligned nanotubes with fiber axial axis helped to transfer the applied load by strain greater than 11.4% [34]. Yesil and co-workers reported a value of electrical resistivity (*ρ*) of poly(ethylene terephthalate) (PET) fibers of 107 Ω·cm, which corresponds to insulator behavior. Meanwhile, PET-based composites fibers with 0.5 wt.% of CNT presented values of 100 Ω·cm characteristic of a semiconductor material [35].

In this work, four series of tailor-made composite yarns are reported. The yarns were made of P(S:AN) and P(S:AN-AA) as polymeric matrices at several compositions and CNTs as reinforcement filler were added. The CNTs were characterized by scanning electron microscopy (SEM) and X-ray diffraction (XRD). Composite polymeric solutions were prepared, and their viscosities were measured. The yarns were electrospun from polymeric solutions and their mechanical and electrical performances were measured as a function of CNTs content. The composite yarns were submitted to a degradation process in saline solution for one month, taking out samples at different times and their elastic modulus was again evaluated.

## 2. Materials and Methods

The synthesis of poly(styrene-co-acrylonitrile) P(S:AN) and poly(styrene-co-acrylonitrile-acrylic acid) P(S:AN-AA) was reported in a previous work [32]. Multiwalled carbon nanotubes (MWCNT) Baytubes C150 P were obtained from Bayer Materials Science, Leverkusen, Germany. Ethanol (Alquimia Mexicana, Mexico City, Mexico), deionized water (Meyer, Mexico City, Mexico), buffer solution (Phosphate), pH 7 (J. T. Baker, Mexico City, Mexico), and 3-(4,5-dimethylthiazol-2-yl)-2,5-diphenyltetrazolium bromide (MTT) (Sigma Chemical, Burlington, MA, USA) were used as reactive. A549 human epithelial cell line, obtained from American Type Culture Collection (ATCC^®^ CCL-185, Manassas, VA, USA) and maintained in F12 culture medium supplemented with fetal bovine serum at 10% (Hyclone Laboratories Inc. Logan, UT, USA), was used for cytotoxicity experiments. All materials were used without further purification.

### 2.1. Carbon Nanotubes Characterization

#### 2.1.1. X-ray Diffraction (XRD)

The crystalline structure of CNTs was identified by X-ray diffraction using a Bruker D8 Focus (Bruker, Madison, WI, USA), with high-intensity monochromatic Cu Kα radiation (λ = 1.5418 Å), operating at 2 ≤ *2θ*/° ≤ 120 and scan rate of 2°/min.

#### 2.1.2. Scanning Electron Microscopy (SEM)

The morphology, structure, and diameter distribution of CNTs were observed by SEM (JEOL JSM-7800F, Tokyo, Japan) at 30 kV and a working distance of 3.0 mm in USD mode. The CNTs for SEM analysis were dispersed with ethanol using an ultrasonic bath for 20 min. A drop was deposited on a copper grid and dried at room temperature. Statistical analysis measurements were performed manually using ImageJ ver. 1.52a (Research Services Branch, NIMH, Bethesda, MD, USA). Before starting measurements, every SEM image was calibrated using the scale bar. In total, 268 measurements on data were made.

### 2.2. Electrospinning Process

#### 2.2.1. Preparation of Composite Solutions

The CNT contents for all composite solutions were 0.5 wt.% and 1.0 wt.% with respect to the polymer weight. A sample of 0.005 g or 0.01 g of CNTs was weighed and well dispersed in 10 g of N,N- dimethylformamide (dielectric constant of *ε* = 37.66) by sonication for 12 h at room temperature [36]. Then, 1 g of polymer was weighed and dissolved in the solution by mechanical stirring for 12 h at 40 °C.

#### 2.2.2. Rheological Properties

The rheological properties of the composite solutions were measured using a Modular Compact Rheometer (MCR 502, Anton Paar, Graz, Austria) in rotation mode. A 1.5 mL sample of spinning solution was deposited on the base of the rheometer and the plane plate geometry (25 mm diameter, 0°) was placed 1 mm above the base. Viscosity measurements were taken from 30 to 120 °C at a scanning rate of 2°/min.

#### 2.2.3. Fiber Fabrication

The polymeric solutions were placed in a 5 mL glass syringe connected to a 21 G needle through PTFE tubing. The needle and the collector were connected to a high voltage power supply. The applied voltage was set at 16 kV. The solutions were fed at 2.5 mL/h using a syringe pump. Electrospun fibers or yarns were collected for 10 min in a blade collector, which was placed 10 cm from the needle tip. The obtained fibers were dried for 1 h at 60 °C. All solutions were electrospun under the same conditions.

### 2.3. Fiber Characterization

#### 2.3.1. Scanning Electron Microscopy (SEM)

The morphology and diameter of fibers and yarns were observed by (SEM JEOL JSM-7800F, Tokyo, Japan) at 30 kV and a working distance of 3.0 mm, USD mode. The samples were dried at 60 °C and coated with gold by sputtering.

#### 2.3.2. Mechanical Properties

A texture analyzer (TA.X2i, Stable Micro Systems, Waverley, UK) was used to measure the elasticity modulus of composite yarns. Samples of 6 cm in length were tested by quadruplicate at a speed of 1 mm/s, using a 25 kg load cell and a sensitivity of 0.10 N.

#### 2.3.3. Electrical Properties

A DC Probe Station (MSTECH 550, MSTECH, Gyeonggi, Korea) was used to analyze the current-voltage ratio in the composite yarns. The tips of the yarns were covered with gold by sputtering to be used as contacts. Yarns of 4 cm length were moistened into 10 mL of saline solution and evaluated at pH 7 using a buffer solution. The yarns were measured from 0 to 20 V. The experiments were performed by triplicate at room temperature.

#### 2.3.4. Cytotoxicity Assay

The cytotoxicity of all-polymer yarns was evaluated in the A549 epithelial cell line using the alamar blue microcolorimetric assay. The A549 cell line was maintained in F12 medium supplemented with 10% fetal bovine serum (FBS) at 37 °C in a 5 v.% CO_2_ atmosphere. Copolymer yarns samples were cut into 1 cm fragments. Then, yarns fragments were sterilized with ethanol for 2 h and exposed to UV light for 30 min.

To prepare confluent monolayers of the A549 epithelial cell line, 80,000 cells/well were added to 48-wells plate. Cells were incubated for 24 h to allow cell adherence. Sterilized yarn samples were placed in each well. Some wells were kept without yarns as a viability control and other wells were added with 80 µg/mL ursolic acid as death control. The plate was incubated for 24 h at 37 °C in a 5 v.% CO_2_ atmosphere. Then, 100 µL of alamar blue (AbD Serotec) were added to each well, and the plate was incubated again until the viability controls were turned pink. Finally, relative fluorescence units (RFU) were measured in SpectraMax M3 Microplate reader (Molecular Devices, CA, USA). The RFU of viability control wells was taken as 100% cell viability.

## 3. Results and Discussion

Four series of composite yarns were obtained by electrospinning technique. The composite materials were prepared by mechanical dispersion of CNTs in a polymer matrix. The polymeric matrices with different monomer ratios were synthesized by means of emulsion polymerization techniques in a semicontinuous power feed process: (i) poly(styrene-co-acrylonitrile) and (ii) poly(styrene-co-acrylonitrile-acrylic acid) [32]. The four compositions and the code names of the composites are shown in Table 1. The structure and purity of CNTs were characterized before dispersing and mixing them into the polymeric matrix.

### 3.1. Carbon Nanotubes (CNT) Characterization

The crystallographic structure of CNTs was analyzed by powder X-ray diffraction (Figure 1). The diffraction pattern showed two characteristic reflections at 25.85° (JCPDS: 96-101-1061) and 42.77° (JCPDS: 41-1487) corresponding to the graphite structure [37,38]. The intensity of the peak in the (002) plane indicated a high crystallinity degree. Cell parameters were calculated and the results are summarized in Table 2 [39].

Morphology, size distribution and length of carbon nanotubes were observed by SEM. The CNTs micrographs at several magnifications are presented in Figure 2. The CNTs show a cylindrical shape without surface impurities such as amorphous carbon.

Figure 2b presents a size distribution of CNTs with diameters up to 20 nm and lengths up to 3 µm. A statistical analysis of MWCNT diameter distribution from SEM image (Figure 2d) was made. The obtained data were grouped into nine classes, resulting in a class width of 3.6 nm. The smallest data were 3.3 nm and the largest data 36 nm. After performing the statistical calculations for grouped data, an average of 12 nm and a standard deviation of 6.2 nm were found, and the mode or diameter that most frequently repeated was 8.8 nm. The diameter distribution as histogram is presented in Figure 3. Figure 2a shows some agglomerations of CNTs, attributed to Van der Waals forces, due to nanotube-nanotube interactions. It was possible to obtain isolated CNTs after being dispersed in ethanol by mechanical stirring and sonication [40].

### 3.2. Composite Polymeric Solutions

This work focused only on the dispersion techniques which could optimally preserve the intrinsic electronic and mechanical properties of CNTs. Composite polymeric solutions of each material were prepared according to Table 1. Carbon nanotubes were well dispersed in DMF using a sonication tip and mechanical stirring. After that, the polymer was added and mechanically stirred until dissolution obtaining a homogeneous solution. The viscosity (η) of composite solutions was measured with an MCR 502 rheometer from 30 to 120 °C with a scanning rate of 1 °C/min. The measurements were made in triplicate. Figure 4 shows the viscosity as a function of temperature of two composite solutions at 10 wt.% with 0:100 wt.%:wt.% for polymeric matrix P(S:AN) and CNTs at 0.5 wt.% and 1.0 wt.% as an example, because all materials showed a similar behavior.

The results showed that the viscosity in the range of 30 < *T*/°C < 75 decreased slowly with the temperature. For example, polymers with 0.5 wt.% CNTs reached viscosity values almost constant of 11.32 Pa·s (±2 Pa·s) until 70 °C, while materials containing 1.0 wt.% CNTs presented values of 12.36 Pa·s (±3 Pa·s) until 75 °C. These viscosity values were higher than viscosity of polymers without CNTs, 10.5 Pa·s (±2 Pa·s) described in a previous work [32]. It has reported that polymeric solutions can be electrospinnable at range of viscosity between 1 < *η*/Pa·s < 15 [41,42]. Below this value, the electrospinning becomes an electrospray technique and if this value is higher, there is entanglement between the polymeric chains and the CNTs, and therefore the fibers are not produced [43]. After these temperatures values, an abrupt increment of viscosity was observed for both solutions, which is attributed to the glass transition temperature (*Tg*) range of the composite polymer. For polymers containing 0.5 wt.% CNTs, the range is 74.93 < *T*/°C < 111.04, while for materials with 1.0 wt.% the found range was 81.82 < *T*/°C < 102.24. After this, the viscosity value approaches until a steady value at 2 kPa.

In a previous work, viscosity analysis and average glass transition range were reported for polymeric solutions; P(S:AN) and P(S:AN-AA) without embedded carbon nanotubes, where *Tg* for P(S:AN) 0:100 was 91.03 °C at ±2 Pa·s [32]. The *Tg* average for bulk PAN was reported by the bibliography to be 90 °C at ±5 Pa·s [44]. Figure 5 shows a comparison between these previously obtained *Tg* and PAN with different CNTs contents: with a content of 0.5 wt.% (92.13 °C at ±10 Pa·s) and 1.0 wt.% (93.01 °C at ±18 Pa·s). That means the glass-transition temperature of the composite increases with the amount of CNTs into the polymeric matrix. The observed increment in *Tg* suggests that CNTs restrict the alignment and movement of the polymeric chains [45].

### 3.3. Fiber Morphology

Four series of composites yarns (SAN0.5, SAN1.0, SAN/AA0.5, and SAN/AA1.0) were electrospun from composite polymeric solutions through the electrospinning technique. The fabrication parameters were: applied voltage 16 kV, working distance 10 cm and feed rate 2.5 mL/h. All fibers were deposited overlapped and punctually entangled on a blades collector, resulting in a yarn-type structure. Figure 6 shows two photographs of obtained composite polymeric yarns SAN/AA1.0 and SAN1.0 as an example. All yarns were measured with a Vernier, founding an average length of 20 ± 3 cm and an average diameter 0.5 ± 0.2 mm.

Figure 7 presents micrographs of composite yarns P(S:AN) 20:80 wt.%:wt.% with 0.5 and 1.0 wt.% CNTs content as an example, since all polymeric concentrations presented similar physical properties. A clear difference in morphology as a function of CNTs content can be appreciated between the yarns, left column (0.5 wt.%) and right column (1.0 wt.%). Figure 7a–c show the presence of CNTs embedded in the polymer matrix in contact with each other and aligned with the axial axis, attributed to the electrostatic stretching of the polymeric solution composed of the electrospinning technique. It has been reported that this arrangement of nanotubes could allow improving the load transfer applied to the yarns and their conductivity [46]. On the other hand, the Figure 7d–f present the yarns with 1.0 wt.% CNTs. The micrographs show that the further addition of nanotubes produces agglomerations, but also zones free of them. Although the dispersion process by sonication and stirring of CNT was completed, when the voltage was applied in the electrospinning process, nanotube–nanotube interactions as Van der Waals forces were stronger than interactions produced by polymer–nanotube, producing the formation of agglomerates. This produces a heterogeneous yarn in which a breaking point and a lack of tensile strength could be found.

### 3.4. Elastic Modulus Composite Yarn Analysis

Stress force as a function of strain was measured on a TA.X2i texturometer for all polymeric yarn series. The mechanical properties of series P(S:AN)—0.5 wt.% and P(S:AN)—1.0 wt.% wt were compared, while P(S:AN-AA)—0.5 wt.% and P(S:AN-AA)—1.0 wt.%, were also contrasted to study the effect of CNTs content on the elastic modulus of yarns with the same polymeric matrix.

The tensile strength behavior of the yarns is shown in Figure 8. The results show an increment of the tensile strength when the acrylonitrile concentration increases into polymeric yarns. The obtained maximum strain corresponds to yarns with a PAN content of 100 wt.%, reaching values of 13 MPa, 17 MPa, 15 MPa, and 26.5 MPa for 1SAN0.5, 1SAN/AA0.5, 1SAN1.0, and 1SAN/AA1.0, respectively. These results present lower values than those reported for P(S:AN) yarns without embedded carbon nanotubes (37 MPa) in a previous work [32]. This can be attributed to the lack of homogeneous orientation of CNTs along the yarns. That is, as the yarns transform into an organized structure, they become more sensitive to the minimum cracks initiated by CNTs bundles due to overloading of the system. It is well known that on PMCs, polymers chains get adsorbed to the charged filler surfaces via interactions as hydrogen bonding and ionic attractions, forming a crosslinked network and leading to enhanced mechanical deformation. However, in the case of CNT at 1 wt.% content, the homogenous phase is disrupted by the aggregation of nanotubes. This means that the transfer charge between the matrix and the carbon nanotubes by polymer–nanotubes interactions is minimized by the presence of agglomerations along the yarn, avoiding the sliding of the polymer chains. Similar results on reduction of mechanical properties were reported by Lisuzzo, where a high concentration of halloysite nanoclays into Mater-Bi matrix caused aggregations and decreased the ultimate elongation of bioplastic films [47].

Figure 8 also shows three characteristic zones for stress–strain curves: the elastic zone, the plasticity zone, and the rupture zone [48]. The elastic zone corresponds to the initial lineal behavior with a defined slope, where the strain is directly proportional to the applied stress and the value of the slope is the elastic modulus. After this yield point, the plasticity zone initiates and ends until break point.

The energy stored up to breaking of the composite yarns during the tensile experiments is also influenced by CNTs content. Figure 9a shows the stored energy for P(S:AN) matrix and Figure 9b for P(S:AN-AA) matrix, where is clearly observed a higher energy stored in both matrices with a 1 wt.% of carbon nanotubes. The interactions between the great surface area of the nanotubes and the polymer matrix facilitate the development of constrained regions with high stiffness, transferring the stress applied on polymer yarn to the reinforcement phase.

The elastic modulus of composite yarns as a function of the polymeric composition is shown in Figure 10a for P(S:AN) at different CNTs contents, while Figure 10b shows results for the P(S:AN-AA) at the same CNTs concentrations. Experiments were carried out by triplicate and error bars indicate standard deviation.

The curves present an increment in the elastic modulus as the concentration of acrylonitrile increases in the polymeric material. This behavior is the same for P(S:AN) and P(S:AN-AA) matrices reaching values up to 27 MPa and 16.5 MPa, respectively. These elastic moduli are close to the maximum values reported for yarns without carbon nanotubes reported in a previous work (30 MPa and 17 MPa) [32]. It has been reported that the elastic modulus of polymeric composite yarns can be taken as a combination of elastic behaviors of the polymeric matrix and the CNTs (orientation and length) [25]. Therefore, this could mean that the content of CNTs can change the reinforcement behavior in the whole yarn. According to results, polymeric material has a greater influence on this property than the CNTs content.

For both series of polymeric matrices, the highest elastic modulus values were found for those with a CNTs content of 0.5 wt.%. This behavior is attributed to the homogenous dispersion and axial alignment of CNTs in the polymeric matrix before and during electrospinning process. At this concentration, the nanotubes were homogeneously dispersed in the polymer matrix and took a very long time to re-aggregate by Brownian motion [23]. However, when 1.0 wt.% of CNTs solution was electrically charged in electrospinning process, the Brownian motion was accelerated by the temperature increment in the solution, resulting in segregation of phases: polymeric matrix and CNTs agglomerations [46]. This was confirmed by images of SEM where areas with CNTs bundle zones and isolated nanotubes were observed (Figure 7). The presence of isolated and aggregated CNT has a negative impact on interfacial bonding with polymeric matrix. Good interfacial bonding is essential to ensure efficient load transfer from the polymeric matrix to the CNTs that helps to reduce stress concentration along the composite yarn [49].

All polymeric composite yarns were submitted to a degradation process in saline solution, where elastic modulus was measured again. The samples were cut into 6 cm lengths and moistened in saline solution at pH 7 for a month. The samples were removed from the solution after one, two, three and four weeks. They were washed with deionized water and dried in an oven.

Figure 11 shows the elastic modulus as a function of polymeric composition of the composite yarns of all series after the degradation process. Experiments were carried out in triplicate and error bars indicate standard deviation. In general, for all series high values of modulus were reached during the first week for the materials with the highest concentration of acrylonitrile. Specially, the highest modulus belongs to series of P(S:AN) with 0.5 wt.% CNTs and P(S:AN) of 1.0 wt.% CNTs with values of 27 MPa and 31 MPa, respectively.

In contrast, the lowest values (2 MPa and 1 MPa) were observed to series P(S:AN-AA) of 0.5 and 1.0 wt.% CNTs with a monomeric concentration of 50:50 wt.%:wt.% for the third week. In all cases, the reported elastic modulus values for the fourth week (average 14 ± 4 MPa) were higher than those reached in the third week. This behavior is attributed to the exchange of Na^+^ ions and functional groups of polymeric chain, allowing the strengthening of the structure. This result is consistent with that reported in a previous work [32].

### 3.5. Electrical Behavior of Composite Yarns

The electrical behavior of the yarns was evaluated as a function of CNTs content. The results show that the dispersion and axial alignment of CNTs in the polymeric matrix were the more important factors on the conductivity of yarns rather than the concentration of CNTs.

The electrical behavior of the yarns with agglomerations (P(S:AN) 50:50 wt.%:wt.%, 1.0 wt.% of CNTs) and aligned CNTs (P(S:AN) 50:50 wt.%:wt.%, 0.5 wt.% of CNTs) is shown in Figure 12. Both experiments with a polymeric matrix of P(S:AN) at 50:50 show an increment of current as voltage increases, characteristic of semiconductor behavior. Moreover, it is evident that the yarn with aligned CNTs (Figure 12a) reached the higher current value, 1.8 mA, meanwhile a value almost 1000 times lower corresponds to the yarn with the presence of agglomerations or clusters (1.5 µA). This significant current drop can be attributed to the CNTs agglomerations that can dissipate the electrical current into the saline solution and/or to the polymeric matrix because several tips of the nanotubes are outside the matrix to allow electrons to leak out. However, these electrical properties were 33% better than those reported for polymeric yarns without CNTs [32].

### 3.6. Cytotoxicity Effect of Composite Yarns on Epithelial Cells

The cytotoxicity evaluation of four series of polymeric composites demonstrated that a gradual increment of CNTs in P(S:AN) and P(S:AN-AA) yarns caused a negative effect on cell viability, where the highest cytotoxicity values were found for P(S:AN) and P(S:AN-AA) 50:50 wt.%:wt.% and 50:50-1 wt.%:wt.%-wt.% yarns (1.0 wt.% of CNTs) (Figure 13).

The assay was performed using microcolorimetric alamar blue assay. As death control was used 80 µg/mL ursolic acid, which showed 92% of death. Results are presented as the means and standard deviation of two independent experiments with two replicates each one. ** Statistically significant difference (*p* < 0.05). Two-way ANOVA with Tukey’s post-test.

**Figure 13 polymers-13-03655-f013:**
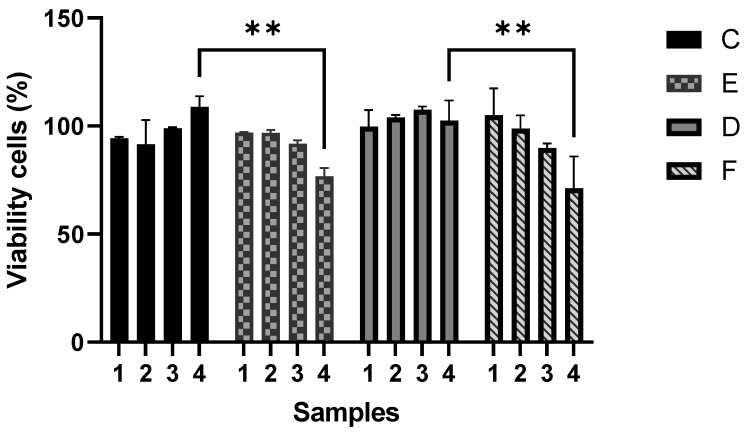
Cytotoxic effect of SAN0.5, SAN1.0, SAN/AA0.5 and SAN/AA1.0 composite series on A549 epithelial cell line.

## 4. Conclusions

As a general result, the elastic modulus of PMCs yarns electrospun with 0.5 wt.% content of CNTs increased almost 40% (27 MPa) compared with those with 1.0 wt.%. This was attributed to a better CNTs orientation and nanotube–polymer interaction, which allows to transfer the load along the obtained yarn with lower concentrations. This was corroborated by SEM micrographs. Electrical properties indicated that the axial alignment of CNTs in the composite yarns plays a key role increasing the electrical current up to 1.8 mA, which was a 33% better than electrical results reported for pristine polymers. According to these results, we can conclude that the most promising performance was achieved by a small CNTs content (≤0.5 wt.%) in poly(styrene-co-acrylonitrile), which might be considered for a possible application as artificial muscles.

## Figures and Tables

**Figure 1 polymers-13-03655-f001:**
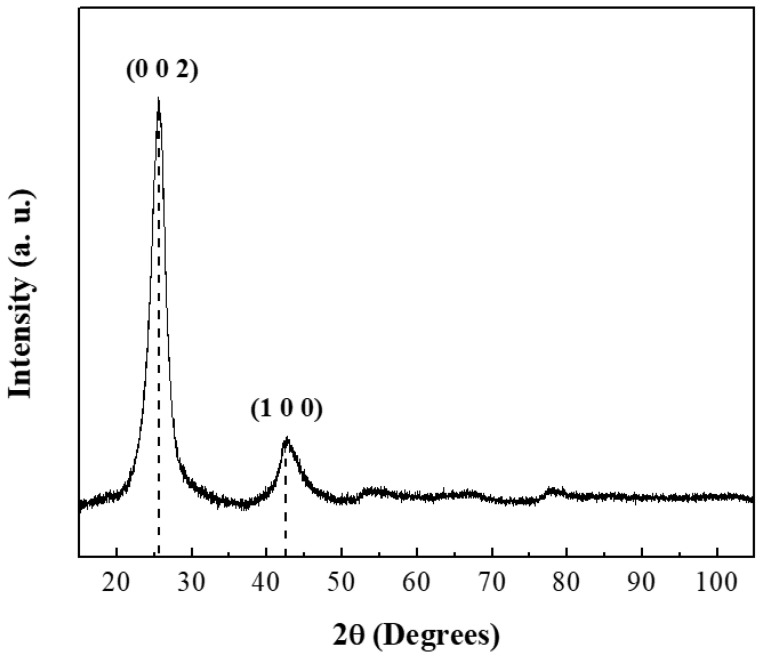
XRD pattern of CNTs.

**Figure 2 polymers-13-03655-f002:**
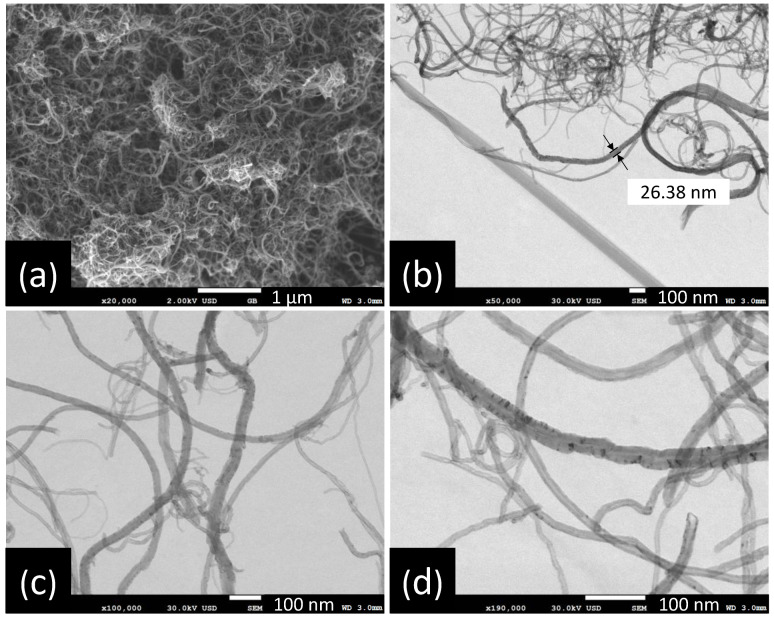
SEM micrographs of CNTs under different magnifications: (**a**) ×20,000; (**b**) ×50,000; (**c**) ×100,000 and (**d**) ×190,000.

**Figure 3 polymers-13-03655-f003:**
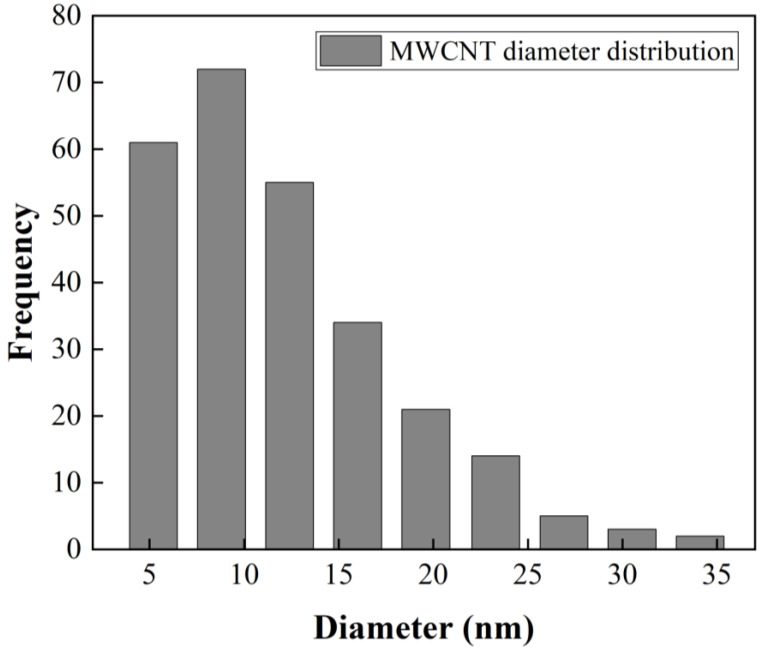
Statistical analysis of MWCNT diameter distribution from SEM image (Figure 2d).

**Figure 4 polymers-13-03655-f004:**
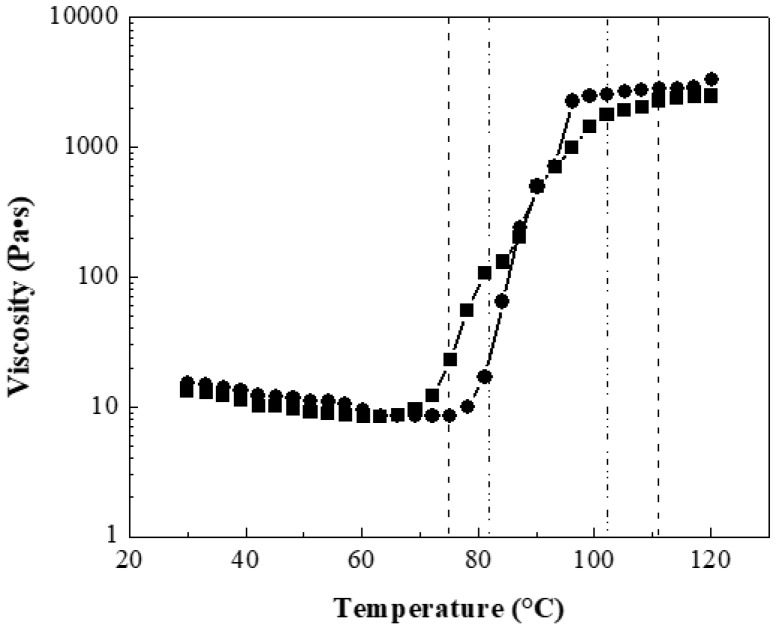
Composite polymeric solutions viscosity of P(S:AN)-CNT 0:100 wt.%:wt.% at two different concentrations of CNTs: 0.5 wt.% (∎) and 1.0 wt.% (●). Glass transition temperature range as function of CNTs: 0.5 wt.% (- -) and 1.0 wt.% (-··-).

**Figure 5 polymers-13-03655-f005:**
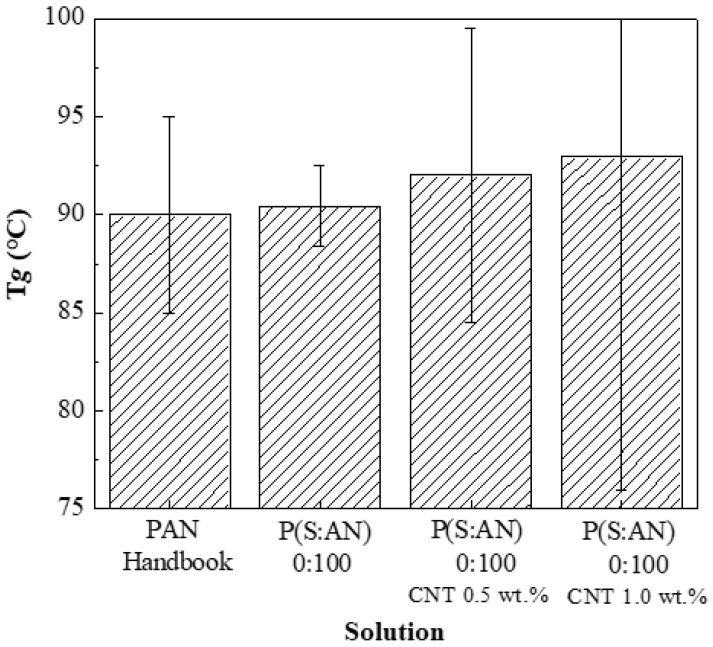
Glass transition temperature of polymer and composite solutions for P(S:AN) 0:100 wt.%:wt.%.

**Figure 6 polymers-13-03655-f006:**
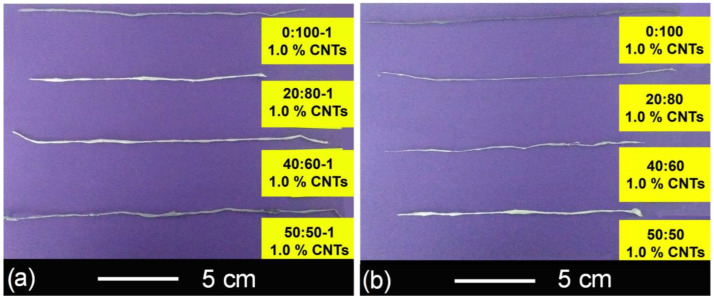
Photographs of composite yarns: (**a**) series SAN/AA1.0 and (**b**) series SAN1.0.

**Figure 7 polymers-13-03655-f007:**
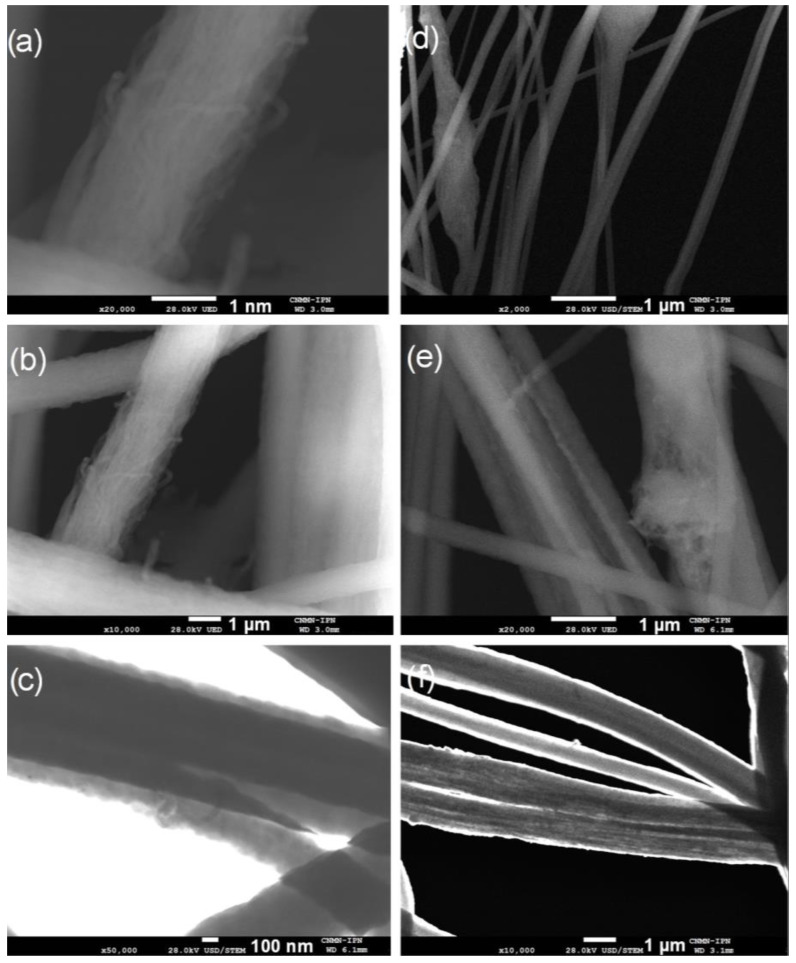
SEM micrographs of P(S:AN) 20:80 wt.%:wt.% yarns with different CNTs contents and amplifications: 0.5 wt.% (**a**) ×20,000 (**b**) ×10,000 (**c**) ×50,000 and 1.0 wt.% (**d**) ×2000 (**e**) ×20,000 (**f**) ×10,000.

**Figure 8 polymers-13-03655-f008:**
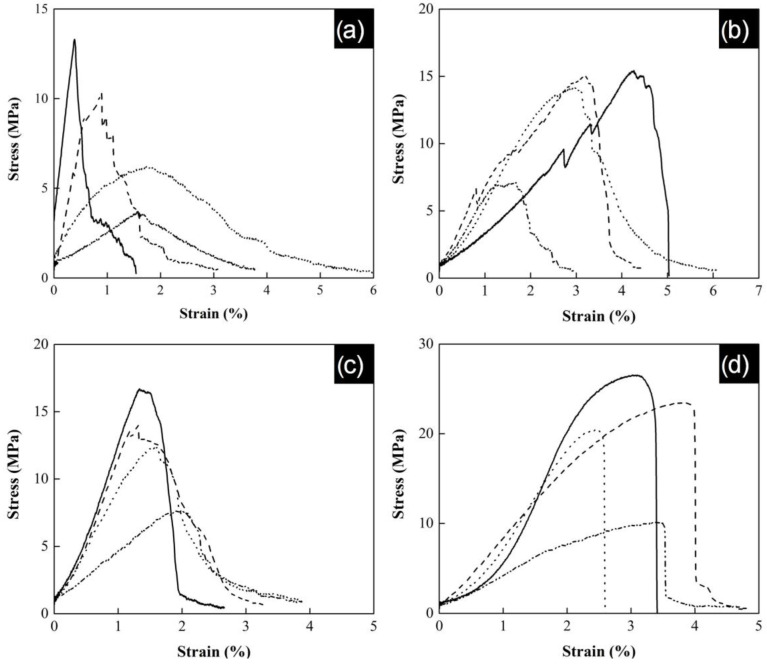
Curve stress as a function of strain, (**a**) series SAN0.5, (**b**) series SAN1.0, (**c**) series SAN/AA0.5 and (**d**) series SAN/AA1.0 yarns: yarn 0:100 (—), yarn 20:80 (- -), yarn 40:60 (··) and yarn 50:50 (-··-).

**Figure 9 polymers-13-03655-f009:**
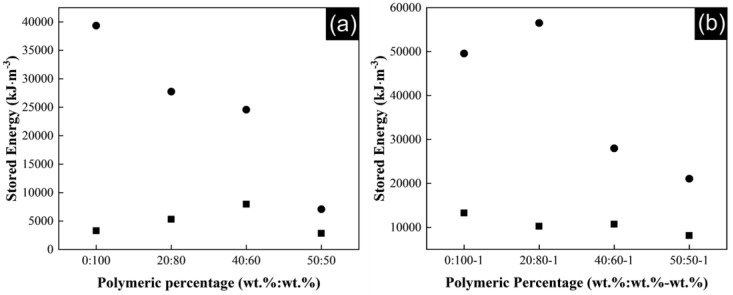
Stored energy up to breaking for composite yarns: (**a**) P(S:AN) and (**b**) P(S:AN-AA), with different CNTs contents: (∎) 0.5 wt.% and (●) 1.0 wt.%. The relative error is 5%.

**Figure 10 polymers-13-03655-f010:**
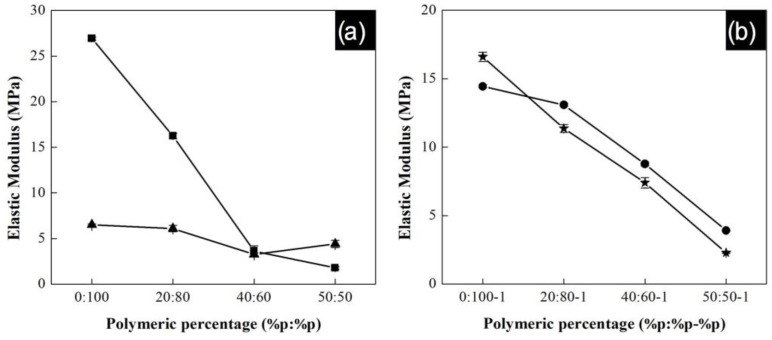
Curve elastic modulus as function of polymeric percentage with different CNTs contents: (**a**) P(S:AN); 0.5 wt.% CNTs (∎); 1.0 wt.% CNTs (▲) and (**b**) P(S:AN-AA); 0.5 wt.% CNTs (●) and 1.0 wt.% CNTs (★). Error bars indicate standard deviation.

**Figure 11 polymers-13-03655-f011:**
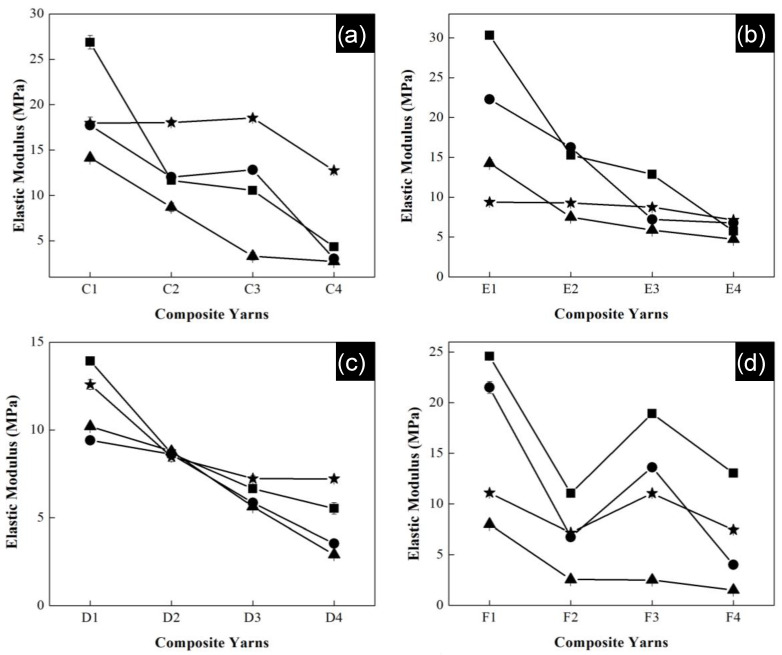
Curve elastic modulus (**a**) series SAN0.5, (**b**) SAN1.0, (**c**) SAN/AA0.5 and (**d**) SAN/AA1.0 composite yarns as a function of polymer percentage with kinetic degradation: 1st week (∎); 2nd week (●); 3rd week (▲) and 4th week (★). Error bars indicate standard deviation.

**Figure 12 polymers-13-03655-f012:**
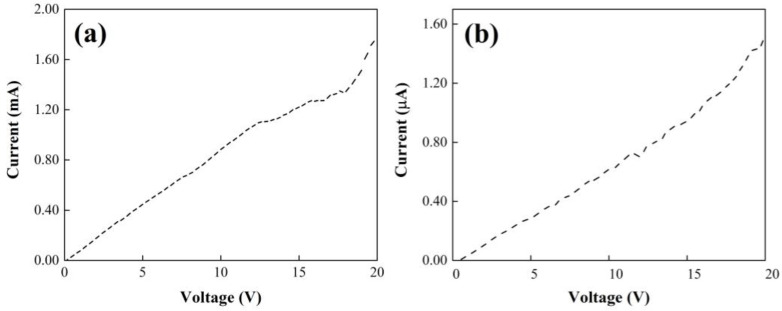
Current as function of voltage for composite yarns of P(S:AN) 50:50 at different CNTs dispersions: (**a**) aligned CNTs, 0.5 wt.% and (**b**) agglomerated CNTs, 1.0 wt.%.

**Table 1 polymers-13-03655-t001:** Composition of composite polymers and code names.

CNT (0.5 wt.%)		CNT (0.5 wt.%)		CNT (1.0 wt.%)		CNT (1.0 wt.%)	
Code	P(S:AN) ^1^	Code	P(S:AN-AA) ^2^	Code	P(S:AN) ^1^	Code	P(S:AN-AA) ^2^
1SAN0.5	0:100	1SAN/AA0.5	0:100-1	1SAN1.0	0:100	1SAN/AA1.0	0:100-1
2SAN0.5	20:80	2SAN/AA0.5	20:80-1	2SAN1.0	20:80	2SAN/AA1.0	20:80-1
3SAN0.5	40:60	3SAN/AA0.5	40:60-1	3SAN1.0	40:60	3SAN/AA1.0	40:60-1
4SAN0.5	50:50	4SAN/AA0.5	50:50-1	4SAN1.0	50:50	4SAN/AA1.0	50:50-1

^1^ P(S:AN); wt.%:wt.%. ^2^ P(S:AN-AA); wt.%:wt.%-wt.%.

**Table 2 polymers-13-03655-t002:** Cell parameters of XRD pattern of CNTs.

2θ	d-Spacing (Å)	(*h k l*)	*β* (FWHM)	*d* (nm)
25.85	3.38815	(0 0 2)	1.245	6.456

## Data Availability

Not applicable.

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
