# Peer review of "Influence of Carbon Nanotubes Concentration on Mechanical and Electrical Properties of Poly(styrene-co-acrylonitrile) Composite Yarns Electrospun"

_polymers, 2021, doi:10.3390/polym13213655_

Round 1

Reviewer 1 Report

The paper is focused on the mechanical and electrical characterization of PMCs yarns electrospun filled with carbon nanotubes. The topic falls within the scope of the journal. The paper can be published after the following revisions:

  • I suggest to conduct a statistical analysis on from SEM images in Figure 2 to determine the size distributions for the diameters of CNTs. The sizes distributions could be reported as histograms in additional figure.
  • Figure 7. Is it possible to determine the stored energy up to breaking from the intregration of the stress vs stain curves.
  • Lines 301-303. The authors stated “This means that the transfer charge between the matrix and the carbon nanotubes is minimized by the presence of agglomerations along the yarn, even if the rest of the CNTs are aligned”. Similar observations were detected for polymers filled with clay nanotubes as reported in recent articles [https://doi.org/10.1016/j.clay.2019.105416] and reviews [DOI: 10.1039/d0tb01865a]. This consideration should be added.
  • Errors bars for the elastic modulus data should clearly reported in Figures 8,9. The authors should explain the mathematical details for the error of the elastic modulus.

Reviewer 2 Report

The abstract has no specific aim, it should be modified.

The sample codes are difficult to follow, should be simplified.

Dispersion should be discussed in detail, from morphology, aggregates were visible. Authors should comment on that.

Can author comment on the interactions? Filler to polymers, filler-to-filler.

Conclusion should be exclusive to only main findings.

Round 2

Reviewer 1 Report

The MS was improved according to the reviewers' suggestions. I recommend its publication in the current form.

Reviewer 2 Report

Authors have answered all the comments.